# Chitosan/Phosphate Rock-Derived Natural Polymeric Composite to Sequester Divalent Copper Ions from Water

**DOI:** 10.3390/nano11082028

**Published:** 2021-08-09

**Authors:** Rachid El Kaim Billah, Moonis Ali Khan, Saikh Mohammad Wabaidur, Byong-Hun Jeon, Amira AM, Hicham Majdoubi, Younesse Haddaji, Mahfoud Agunaou, Abdessadik Soufiane

**Affiliations:** 1Laboratory of Coordination and Analytical Chemistry, Department of Chemistry, Faculty of Sciences, University of Chouaib Doukkali, El Jadida 24000, Morocco; rachidelkaimbillah@gmail.com (R.E.K.B.); m.agunaou@gmail.com (M.A.); abdessadiksoufiane@gmail.com (A.S.); 2Chemistry Department, College of Science, King Saud University, Riyadh 11451, Saudi Arabia; swabaidur@ksu.edu.sa; 3Department of Earth Resources and Environmental Engineering, Hanyang University, Seoul 04763, Korea; bhjeon@hanyang.ac.kr; 4Laboratory of Analytical Chemistry and Physico-Chemistry of Materials, Department of Chemistry, Faculty of Sciences Ben M’Sik, University of Hassan II-Casablanca, Casablanca 21100, Morocco; Amamira6@gmail.com (A.A.); hichammajdoubi.hm@gmail.com (H.M.); 5Laboratory of Engineering and Materials, Department of Chemistry, Faculty of Sciences Ben M’Sik, University of Hassan II-Casablanca, Casablanca 21100, Morocco; ys.haddaji@gmail.com

**Keywords:** chitosan, calcium fluorophosphate, copper, adsorption, regeneration

## Abstract

Herein, a chitosan (CH) and fluroapatite (TNP) based CH-TNP composite was synthesized by utilizing seafood waste and phosphate rock and was tested for divalent copper (Cu(II)) adsorptive removal from water. The XRD and FT-IR data affirmed the formation of a CH-TNP composite, while BET analysis showed that the surface area of the CH-TNP composite (35.5 m^2^/g) was twice that of CH (16.7 m^2^/g). Mechanistically, electrostatic, van der Waals, and co-ordinate interactions were primarily responsible for the binding of Cu(II) with the CH-TNP composite. The maximum Cu(II) uptake of both CH and CH-TNP composite was recorded in the pH range 3–4. Monolayer Cu(II) coverage over both CH and CH-TNP surfaces was confirmed by the fitting of adsorption data to a Langmuir isotherm model. The chemical nature of the adsorption process was confirmed by the fitting of a pseudo-second-order kinetic model to adsorption data. About 82% of Cu(II) from saturated CH-TNP was recovered by 0.5 M NaOH. A significant drop in Cu(II) uptake was observed after four consecutive regeneration cycles. The co-existing ions (in binary and ternary systems) significantly reduced the Cu(II) removal efficacy of CH-TNP.

## 1. Introduction

Heavy metals are non-biodegradable. Beyond maximum allowable levels, they are considered toxic to living communities. They have a tendency to accumulate in living organisms. Copper (Cu(II)) is an essential nutrient for brain growth and the nervous system. At higher concentrations, it has the potential to kill microorganisms (especially bacteria and fungi) [1]. However, high Cu(II) levels could modulate hemoglobin synthesis, create gastrointestinal infection, cause lung cancer, and in worst case scenarios, lead to death [2]. Mining is the major carrier of Cu(II) to surface and subsurface water [3]. Additionally, industrial effluents from metal extraction and electroplating plants, pulp and paper mills, printing presses, and fertilizer plants discharge Cu(II) to ground water [4]. According to the World Health Organization’s (WHO’s) standard, the maximum acceptable concentration of Cu(II) in drinking water is 2.0 mg/L [5]. Several methods, such as membrane filtration [6], reverse osmosis [7], electrolysis [8], precipitation [9], nanofiltration [10], coagulation/co-precipitation [11], ion-exchange [12], liquid-liquid extraction [13], and adsorption [14], have been engineered to treat Cu(II) ions enriched waste effluents.

Among the aforementioned processes, adsorption is recognized as a simple and cost-effective strategy to remove heavy metals from aqueous environments [15]. Activated carbon is the most extensively studied adsorbent for heavy metal removal. However, the high regeneration cost limits its utilization for practical applications. Various non-conventional adsorbents, such as zeolite [16], chitosan [17], hematite and goethite [18], montmorillonite [19], hydroxyapatite [20], and alginate [21], have been studied for the removal of Cu(II) from water. Among them, chitosan (CH, (C_6_H_11_NO_4_)_n_) seems to be a promising adsorbent due to its biodegradable and biocompatible features. In addition, this material has an ample number of free amine and hydroxide groups in its matrix from a biopolymer chain of *N*-acetylglucosamine [22]. Thus, it can aid in adsorbing a large number of metal ions through chelation. The chelating efficiency of CH is about five to six-folds greater than its precursor chitin [23]. However, the metal adsorption capability of CH varies greatly with the pH. At lower pH, CH swells and disintegrates due to the protonation of amino groups [24]. This restricts its use in commercial water treatment application.

Compositing could be an effective strategy to retain CH adsorption capability while controlling its solubility, swelling, and agglomeration. Chemical cross-linking has been shown to enhance CH stability under acidic environments and reduces its hydrophobicity [25]. Similarly, CH has been coupled with manganese ferrite biochar [26], mesoporous silica (MCM-48) [27], and amidoxime [28] to demonstrate high adsorption capabilities along with improved adsorbent stabilities. Functional beads were developed by using water soluble agar aldehyde and CH for the removal amido black dye [29]. Phosphorus-containing composites have received considerable attention due to their excellent chelating (due to the Lewis base properties of phosphate group) and metal-adhesion properties [30]. Magnetic phosphorylated CH composited was developed for the highly efficient and selective removal of bivalent lead ions from water [31]. Calcium phosphate, particularly the apatitic-type which forms the majority of phosphate ores, has a wide variety of physico-chemical and textural properties [32]. The presence of minerals, e.g., calcite and dolomite, in sedimentary apatite is highly beneficial for phosphorus retention due to the excellent dissolution of these carbonate minerals which results an increase in the pH of the medium and consequently in the Ca^2+^ ion concentration in solution, favoring the calcium phosphate precipitation [33].

In the present investigation, a CH and fluorapatite (TNP) based CH-TNP composite was synthesized using seafood waste and natural occurring phosphate rock for heavy metal adsorption. Structural and chemical properties of the synthesized CH-TNP composite were studied using microscopic and spectroscopic studies, respectively. Batch experiments were conducted to examine the adsorption and regeneration ability of CH-TNP composite. In addition, studied experimental parameters were optimized. An adsorption mechanism is postulated.

## 2. Experimental

### 2.1. Materials

The natural phosphate sample used in this study was collected from Khouribga province, Morocco. This mining city is considered to be the most important phosphate production area in the world with estimated three-quarters of the global phosphate reserves [34]. Analytical reagent (A.R) grade copper (II) nitrate trihydrate (Cu(NO_3_)_2_·3H_2_O), sodium hydroxide (NaOH), hydrochloric acid (HCl), sodium chloride (NaCl), sulfuric acid (H_2_SO_4_), ethanol (C_2_H_5_OH), magnesium nitrate (Mg(NO_3_)_2_), sodium nitrate (NaNO_3_), and potassium nitrate (KNO_3_) were obtained from Sigma-Aldrich, Germany. All solutions were prepared using deionized (D.I.) water from a Milli-Q water purification system (Millipore, Burlington, MA, USA).

### 2.2. Preparation of CH-TNP Composite

#### 2.2.1. Preparation of Chitosan (CH)

Since CH was a derivative of chitin, we initially extracted chitin from crustacean shells in two-steps, which involves demineralization and deproteinization. Initially, the shrimp shells were thoroughly washed with tap water and thereafter left for overnight drying in an oven at 50 °C. Demineralization was performed at room temperature by adding 1 M HCl solution (*m*/*v* 1:15). Then, the mixture was vigorously stirred for 5 h. The resulting product was washed with D.I water neutral pH, filtered, and dried for 24 h at 50 °C. Deproteinization was carried out at 90 °C using 5% NaOH solution (*m*/*v* 1:20) through refluxing (10 h) under continuous stirring condition. The chitin produced was filtered out, washed to neutral pH with D.I water, and thereafter dried at 50 °C for 24 h.

Finally, the CH was obtained by removing acetyl groups through deacetylation. The deacetylation procedure was carried out by refluxing the mixture containing 10 g of chitin and 100 mL of 50% NaOH solution at 100 °C for 6 h in a round bottom flask. The obtained product was washed with D.I water and dried for further use.

#### 2.2.2. Preparation of Fluorapatite (TNP)

The collected phosphate samples were pre-treated in order to remove the impurities associated with phosphate minerals and to have a well-determined general profile and homogeneity. Approximately 20 g of the natural phosphate was added to a 500 mL flask. The ore dissolution reaction was carried out by adding 250 mL of D.I water at 75 °C for 2 h. The sample was filtered, dried overnight at 100 °C, and thereafter calcined for 2 h at 900 °C. The calcined sample was chemically activated with 150 mL 1 N HNO_3_ for 2 h at room temperature under continuous stirring. The resulting filtrate was then neutralized with 150 mL of 25% NH_4_OH solution to prevent the formation of acid phosphate. The precipitate that forms was left to mature under magnetic stirring, and then vacuum filtered, washed with D.I water, and dried in an oven at 100 °C for 24 h.

#### 2.2.3. Preparation of CH-TNP Composite

The composite was prepared by mixing 6 g of CH in 80 mL of 1% (*v*/*v*) CH_3_COOH solution with 2 g of TNP. This mixture was then poured into a PET plastic container and dried at room temperature for 48 h. The dried CH-TNP composite film was peeled off from the PET plastic container and stored for characterization and adsorption studies.

### 2.3. Characterization of CH-TNP Composite

The X-ray diffraction analysis (XRD, Bruker D8 Advance, Billerica, MA, USA) was carried out to examine the XRD patterns of CH, TNP, and CH-TNP composite samples. The chemical functionalities present on CH, TNP, and CH-TNP composite (before and after Cu(II) adsorption) samples were analyzed by Fourier transform infra-red (FT–IR, Perkin Elmer 2000, Waltham, MA, USA) spectroscopy. Scanning electron microscopy (SEM, QUANTA FEG 650, ELECTRON MICROSCOPY, Bellaterra, Spain) was employed to study the surface morphology the samples. ASAP 2020 (Micromeritics, Norcross, GA, USA) surface area analyzer was employed to measure the surface area through BET (Brunnauer–Emmett–Teller) method. Before analysis, the samples were outgassed at 150 °C for 18 h. The thermal stability of the samples was analyzed through thermogravimetric analysis (TGA/DTG, STD Q 600, Artisan Technology Group, Kansas City, MO, USA).

### 2.4. Environmental Remediation Application

#### 2.4.1. Adsorption Studies

The Cu(II) adsorption studies were carried out through batch scale experiments. Various experimental parameters such as effect of contact time (t: 1–60 min), solution pH (pH: 1–7), adsorbent dose (m: 0.02–0.2 g), and initial concentration (C_o_: 20–100 mg/L) were studied by using adsorbate solution volume (V: 50 mL). During the experiments (other than contact time) the solid/solution phases were equilibrated over a shaker at 100 rpm for 60 min at room temperature (~25 °C). At equilibrium, the solid/solution phases were centrifuged (Universal 320R Refrigeree Hettich, Berlin, Germany) at 240 rpm for 30 min and filtered to ensure complete removal of solid adsorbent particles. The final pH of the aliquot samples was measured. The adsorption kinetics study was carried out at C_o_: 100 mg/L. The residual concentration of samples was analyzed by inductively coupled plasma-atomic emission spectrometry (Thermo Jarrell Ash Corporation Atom Scan 16, Williamston, SC, USA).

The uptake at equilibrium (*q_e_*_,_ mg/g), at time (*q_t_*_,_ mg/g), and percentage (%) adsorption were estimated using the equations given below:(1)$$q_{e}=\left(C_{o}-C_{e}\right)\times \frac{V}{m}\;$$
(2)$$q_{t}=\left(C_{o}-C_{t}\right)\times \frac{V}{m}\;$$
(3)$$Adsorption\;\left(\%\right)=\frac{\left(C_{o}-C_{e}\right)}{C_{o}}\times 100\;$$
where, *C_o_*, *C_e_*, and *C_t_* (mg/L) are the concentration of adsorbate at initial, equilibrium, and at time *t*, *V* (L) is the solution volume, and *m* (g) is the mass of adsorbent.

#### 2.4.2. Desorption, Regeneration, and Co-Existing Ions Studies

Similarly, the desorption studies were also carried out through batch mode using 0.5 M HCl, 0.5 M NaOH, and 0.5 M NaCl solutions as eluents. Briefly, 0.1 g of CH-TNP composite was saturated with 50 mL Cu(II) solution of C_o_: 100 mg/L. At equilibrium, the residual solution was separated (as previously mentioned in Section 2.4.1) and its concentration was quantitatively determined by ICP-AES. The Cu(II) saturated CH-TNP composite was washed several times with D.I water to ensure complete removal of unabsorbed Cu(II) ions from its surface. Thereafter, the sample was treated with 50 mL 0.5 M HCl solution. The same procedure was followed elute Cu(II) ions from saturated CH-TNP by other eluents. The efficacy of the adsorbent was tested through regeneration studies. Eluent with maximum Cu(II) ions recovery from saturated CH-TNP was selected for regeneration. In addition, the effect of co-existing ions, such as Mg^2+^, K^+^, and Na^+^ (in binary and ternary systems), on Cu(II) adsorption onto CH-TNP was also studied using their respective nitrate salts (Mg(NO_3_)_2_, NaNO_3_, and KNO_3_).

## 3. Results and Discussion

### 3.1. Characterization of Adsorbent

Figure 1A illustrates the XRD patterns of CH, TNP, and the CH-TNP composite. The CH diffractogram shows characteristic peaks at 10.16° and 21.8° corresponding to a chitosan polymorph [34], which has been confirmed by the JCPDS file no. 039-1894. The absence of peak at 12.0° (characteristic of chitin) clearly indicates the complete conversion of chitin to CH [35]. The diffraction patterns of TNP show multiple peaks which corresponds to hexagonal phases of fluorapatite (JCPDS file no.15-0876) [34]. The XRD pattern of CH-TNP composite showed a predominance of TNP peaks with slight peak broadening. The CH peaks also became less intense with a slight shift in the 2θ values. These features clearly suggest that the interaction between CH and TNP was obviously achieved under the described synthetic conditions.

The FT-IR spectra of CH were in line with previous studies [36,37] (Figure 1B). A strong band appeared between 3000 and 3600 cm^−1^ associated to N-H and O-H vibrations. A peak at 2930 cm^−1^ was allocated to C–H symmetric stretching vibration. Meanwhile, a peak at 1661 cm^−1^ corresponds to residual N-acetyl (C=O stretching of amide I). A band at 1420 cm^−1^ was linked to CH_2_ bending and CH_3_ symmetrical deformations. A band associated with asymmetric stretching of the C–O–C bridge appeared at 1151 cm^−1^, while a band at 1030 cm^−1^ was related to C–O stretching. The FT-IR spectrum of TNP showed cluster of bands characteristic of phosphate (PO_4_^3−^) at 566.9, 602.6, and 1042.8 cm^−1^. Two conjoint weak bands of CO_3_^2−^ appeared at 1455 cm^−1^ [38]. In the CH-TNP composite, the protonation of CH amine functionalities was suggested by the presence of the band, attributed to NH_3_^+^ groups, namely the bending vibration at 1475 cm^−1^. This was obviously conditioned by the formation of electrostatic bonds with TNP. Moreover, a low frequency shift of the amide was observed by a band at 1661 cm^−1^, which indicates the participation of the carbonyl groups of the CH amide groups in hydrogen bonding with hydroxyl groups of TNP.

The SEM images of CH, TNP, and the CH-TNP samples were depicted in Figure 2. The SEM image of CH displayed a homogeneous morphology with smooth surface characteristics (Figure 2A), whereas the TNP sample depicts a rough surface with particles of uneven size and irregular shape (Figure 2B). It also revealed that the TNP particles exist in irregular size and geometry with some fine particles covering the surface of TNP. The surface morphology of the pristine CH-TNP composite (Figure 2C) showed the presence of fluroapatite over the entire surface of the composite due to TNP aggregation at the bottom of the composite film. Figure 2D displayed the complete change in surface morphology of the CH-TNP composite after the adsorption of Cu (II), which suggests the metal ions retention on its surface.

Figure 2E displayed N_2_ adsorption-desorption isotherms of CH, TNP, and CH-TNP composite. All samples showed a Type IV isotherm patterns. This is indicative of monolayer adsorption followed by multilayer formation through capillary condensation [39]. The measured surface areas of the adsorbents were 16.75, 63.3, and 35.5 m^2^/g for CH, TNP, and the CH-TNP composite, respectively. The composite adsorbent had ~2.2 times higher surface area than the parental form of CH. This might be due to the functionalization of TNP over the surface of CH. Since TNP has a comparatively higher surface area than CH, this could provide an increase in surface area by facilitating the formation of the composite adsorbent (Table 1). Importantly, the pore size of the CH-TNP composite falls in the mesoporous range (2–50 nm), which is beneficial for the adsorption.

The thermogravimetric (TG) analysis of CH and CH-TNP composite were carried out in the temperature range of 45–700 °C, as shown in Figure 3A. The weight loss for both CH and CH-TNP composite showed two steps between 125 and 300 °C and 300 and 600 °C, ascribed to the desorption of adsorbed water molecule (dehydration) and the thermal dissociation of CH and dehydroxylation of TNP, respectively [40]. The DTG curves showed the maximum dissociation rate of the polymeric CH-TNP composite occurred at 312 °C, as shown in Figure 3B.

### 3.2. Adsorption Studies

The influence of pH on Cu(II) adsorption onto both CH and CH-TNP composite was investigated by varying the solution pH from 1–7. To study the pH effect, 0.1 g of adsorbents were used to treat 50 mL solutions of Cu(II) (C_o_: 100 mg/L) for 1 h at room temperature. At pH: 1, the Cu(II) adsorption was very low as -NH_2_ groups present on CH and CH-TNP surface tend to be in the protonated –NH_3_^+^ form, resulting in a loss in its chelating ability with Cu(II) ions [41]. In addition, at lower pH, there was a competition between H_3_O^+^ ions and Cu(II) ions to occupy active adsorption site. A sharp increase in Cu(II) adsorption on both CH and the CH-TNP composite was observed between pH 1 and 3, illustrated in Figure 4A. The increase in pH facilitates Cu(II) ions adsorption as the degree of protonation of amino (–NH_2_) groups was reduced and the dissociation level of hydroxyl (–OH) groups increased [42]. This leads to an increase in chelation and electrostatic interactions between both CH and CH-TNP and Cu(II) ions. Thereafter, the Cu(II) uptake in between pH 3 and 4 on CH was almost stagnant, while a slight drop in uptake was observed on CH-TNP composite. At pH > 4, the precipitation of insoluble metal hydroxides was taken place. This shows a decrease in Cu(II) ions removal tendency on both CH and CH-TNP composite. Thus, CH showed a maximum (66%) Cu(II) uptake at pH: 4, while on the CH-TNP composite the maximum (85%) Cu(II) uptake was found at pH 3. The presence of deprotonated PO_4_^3−^ groups along with –NH_2_ and –OH groups on CH-TNP composite aid in comparatively higher Cu(II) ion uptake by changing the electrical properties of CH-TNP composite surface to make it negatively charged [31], thus providing better electrostatic interaction to positively charged Cu(II) ions with CH-TNP composite surface. Based on the results, the identified optimum Cu(II) adsorption pH for CH-TNP composite was pH 3.0.

Figure 4B displayed the contact time plot for Cu(II) ions adsorption on CH and CH-TNP. This study was conducted by equilibrating 50 mL of Cu(II) solutions of C_o_: 100 mg/L with 0.1 g of adsorbents (CH and CH-TNP) at pH 3.0 for fixed time intervals (varied between 0 and 60 min). The solid/solution phase were separated and the residual Cu(II) ion concentration was measured. Rapid Cu(II) uptake during the initial 20 min of contact time due to the availability of a large number of unsaturated active binding sites was observed on both CH and CH-TNP. This was due to external surface adsorption. Thereafter, it became slow as most of the active binding sites were gradually exhausted and the immobilized metal ions proceeded in the interlayer of CH and CH-TNP. This was due to intraparticle diffusion. The maximum Cu(II) ion uptakes on CH and CH-TNP were 63% and 84%, respectively and the recorded equilibration time was 40 min. Higher Cu(II) uptake on CH-TNP shows that the introduction of PO_4_^3−^ groups can improve Cu(II) ions adsorption performance.

Figure 4C illustrated the effect CH and CH-TNP (varied between 10 and 200 mg) dose on the adsorption of Cu(II) ions, with an initial concentration of 100 mg/L (50 mL) at pH 3.0 under room temperature conditions. The increase in dose from 10 to 100 mg results an increase in adsorption capacity, gradually reaching the maximum value at 100 mg for both adsorbent samples. Furthermore, the increase in dose from 100 to 200 mg indicated that there is no appreciable change in the Cu(II) intake on both CH and CH-TNP adsorbents.

### 3.3. Adsorption Modeling

#### 3.3.1. Isotherm Modeling

The Cu(II) adsorption data were evaluated using two different isotherm models namely Langmuir and Freundlich (Appendix A). Table 2 presents the calculated value of correlation coefficient (R^2^). Langmuir model was fitted well to adsorption data compared to the Freundlich model. This suggests monolayer adsorption of Cu(II) onto the prepared composite (CH and CH-TNP). The magnitudes of maximum monolayer adsorption capacities (q_m_) of Cu(II) ions on CH and CH-TNP were 37.73 and 45.87 mg/g, respectively. The obtained maximum adsorption capacities were compared with the previous studies in the literature. It was found that the synthesized composite had comparatively higher adsorption capacity (Table 3 [35,43,44,45,46,47]). The magnitude of Langmuir constant (b) showed comparatively higher binding affinity towards Cu(II) on CH-TNP than CH.

#### 3.3.2. Kinetic Modeling

To determine the kinetic parameters for Cu(II) adsorption on CH-TNP, the pseudo-first-order and pseudo-second-order kinetic models (Appendix A) were applied. The Cu(II) adsorption data on CH and CH-TNP were better fitted to pseudo-second-order kinetic model (based on the higher R^2^ values) (Table 4). These findings were further affirmed by nearer q_e,exp._ and q_e,cal._ values for Cu(II) adsorption on CH-TNP and CH. This implies that Cu(II) adsorption on CH and CH-TNP was a chemical adsorption process rather than an ordinary mass transport [48]. The applicability of pseudo-second order model was also observed during previous studies on CH-based composites for Cu(II) adsorption (Table 3).

#### 3.3.3. Thermodynamics Parameters

The thermodynamic parameters associated to Cu (II) adsorption, namely the standard free energy change (∆G°), the standard enthalpy change (∆H°), and the standard entropy change (∆S°) were calculated [49,50] (Appendix A) and the obtained values are presented in Table 5. The thermodynamic processing of the adsorption data indicates that the values of ∆G° for CH and CH-TNP were negative at all studied temperature ranges, indicating the spontaneous nature of Cu(II) adsorption process. Moreover, the negative ∆G° magnitude confirmed that Cu(II) adsorption on both CH and CH-TNP was thermodynamically feasible. The positive value of ∆S° indicates randomness at the solid–solution interface during Cu(II) adsorption on both CH and CH-TNP. During adsorption, the Cu(II) species displace the adsorbed water molecules and the latter gain more translation entropy than is lost by the former, resulting in a prevalence of randomness in the system [51,52]. The positive value of ∆H° for Cu(II) removal confirms the endothermic nature of the adsorption process on both adsorbents.

### 3.4. Desorption and Regeneration

The adsorption capacity and desorption properties are two vital parameters used to evaluate the efficiency of the adsorbent. Herein, three eluents were chosen based on the literature report such as NaOH, NaCl, and HCl. The concentration of each eluent was set at 0.5 M with a fixed contact time of 16 h. The results showed that only 23.8% of Cu(II) ions were desorbed with 0.5 M NaCl, whereas, 42.5 and 81.6% of Cu(II) ions were desorbed when 0.5 M HCl and NaOH solutions, respectively were used. Based on the results, it was found that 0.5 M NaOH was the best desorbing agent for Cu(II) from CH-TNP composite, illustrated in Figure 5A.

To evaluate the recycling efficiency of CH-TNP, repeated Cu(II) adsorption cycles were investigated and the results are displayed in Figure 5B. The Cu(II) ions removal efficiency was tested for four consecutive regeneration cycles and it was found that the removal efficiencies of > 82.5% could be achieved with CH-TNP composite. Thereafter, a significant drop in Cu(II) removal efficiency to 66.4 and 51.9% for the respective fifth and sixth regeneration cycles was observed. This might be due to the saturated of Cu(II) ions over the CH-TNP composite surface, which occurred via a strong interaction between PO_4_^3−^ groups and Cu(II) ions or material loss during adsorption–desorption cycles. The results confirm that CH-TNP composite can be effectively re-used for five consecutive cycles with sustained efficiency and up to six cycles for practical purposes depending on the concentration of Cu(II) ions.

The economical efficacy of CH-TNP was evaluated through reusability studies, presented in Figure 5B. Stagnant Cu(II) ions removal efficiency of >82.5% was observed for four consecutive regeneration cycles from CH-TNP. Thereafter, a significant drop in Cu(II) removal efficiency to 66.4% and 51.9% during the fifth and sixth respective regeneration cycles was observed. This might due to the accumulation of Cu(II) ions over CH-TNP surface, which occurred via a strong interaction between PO_4_^3−^ groups and chelating Cu(II) ions, resulting in incomplete Cu(II) ion desorption from CH-TNP. This is attributed to a loss of active binding sites on CH-TNP surface. In addition, CH-TNP loss in multiple washing steps during adsorption–desorption cycles might also responsible for a decrease in removal efficiency. The results found that CH-TNP is reusable and efficient for four consecutive regeneration cycles.

### 3.5. Effect of Co-Existing Metal Ions on Cu(II) Adsorption

The affinity of CH-TNP for Cu(II) adsorption in the presence of Mg^2+^, Na^+^, and K^+^ was studied by taking 0.1g of adsorbent and 100 mg/L of each 50 mL metal ion. The adsorption capacity of CH-TNP for Cu(II) in the binary and ternary adsorption systems is presented in Figure 6. The results clearly state that, in binary or ternary metal ion solution systems (containing Mg^2+^, Na^+^, or K^+^), the Cu(II) removal potential of CH-TNP decreased. This may be associated to the metal ions competitiveness for the adsorbent’s active sites [47,48].

### 3.6. Adsorption Mechanism

Based on the obtained results, the mechanism of Cu(II) adsorption was proposed as follows. As illustrated in the Scheme 1, the outer surface of the CH was enriched with and nitrogen (–NH and –NH_2_) and oxygen (–OH and –O–) containing functionalities. In addition, the introduction of PO_4_^3−^ groups on CH surface by compositing with calcined phosphate facilitates Cu(II) ion intake through chelation mechanisms on the CH-TNP composite. All these functionalities present on the CH-TNP surface play an active role in Cu(II) adsorption. The FT-IR spectrum of pristine CH-TNP (Figure 6B) showed a broad band at 3253.28 cm^−1^ and sharp bands at 1528.35 and 1024 cm^−1^. These respective bands were slightly shifted to 3232.07, 1534.51, and 1022.53 cm^−1^ in the Cu(II) saturated CH-TNP spectrum, affirming the involvement of –OH, –NH_2_, and C–O–C functional groups in Cu(II) ion binding over the CH-TNP surface during adsorption. The higher electronegativity of the heteroatoms (O and N) [53] allows the binding of positively charged Cu(II) ions via electrostatic and van der Waals interactions, in line with previous works [48,49]. Moreover, the sharing of a lone pair of electrons from N and O atoms with Cu(II) ions supports the formation of co-ordinate bonding between them. Overall, the higher Cu(II) uptake on the CH-TNP composite can be explained by the availability of excessive oxygen binding sites due to the incorporation of PO_4_^3−^ containing functional groups during compositing process.

## 4. Conclusions

Conclusively, the introduction of PO_4_^3−^ groups over the CH surface led to the formation of a CH-TNP composite using a naturally occurring apatite mineral. The obtained composite exhibits high efficiency for Cu(II) ion adsorption between pH 3 and 4 through electrostatic interaction and chelation effects. The pH optimization studies clearly proved that the solution pH influences Cu(II) adsorption on the CH-TNP composite. However, the CH-TNP composite showed comparatively better Cu(II) adsorption performance than CH, under the same experimental conditions. The Cu(II) uptake on both CH and CH-TNP was rapid for the initial 20 min, attaining equilibrium in 40 min with 63% and 84% Cu(II) adsorption, respectively. Finally, 0.5 M NaOH was identified as a most suitable Cu(II) eluent from the CH-TNP composite and this newly developed composite can be efficiently re-used for up to four consecutive regeneration cycles. In addition, coexisting ions (in binary and ternary-systems) have a profound influence on Cu(II) uptake over CH-TNP composite.

## Data Availability

No data available.

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
