# Peer review of "Chitosan/Phosphate Rock-Derived Natural Polymeric Composite to Sequester Divalent Copper Ions from Water"

_nanomaterials, 2021, doi:10.3390/nano11082028_

Round 1
Reviewer 1 Report
The title is clear. The content is in accord with title.
The manuscript adheres to the journal's standards after revision.
The size of the article is appropriate to the contents.
The authors must underline the major findings of their work and explain how the use of their proposed procedures and materials represent a progress to other similar studies.
The Abstract must be revised.
The key words permit found article in the current registers or indexes.
In the introduction it is clearly described the state of the art of the investigated problem. The references from last years were be cited.
The methods are well described and the equipment and materials have been adequately described.
Please explain the experimental parameters: why contact time was 60 min? pH value? etc.
The tables contain necessary results.
The figures have a good quality.
The Conclusion is been justified sufficiently.
References from last years are presented.
Please provide minimum 2 references from this journal (last years), for demonstrated that manuscript is in Nanomaterials topic.
The literature is sufficiently critical, current, and internationally evaluated.
The paper has the text presented and arranged clearly and concisely.
The paper was written in standard, grammatically correct English, small corrections are necessary.
PLEASE USE TEMPLATE FOR MANUSCRIPT.
Please respect author guide.
Author Contributions: The following statements should be used “Conceptualization, X.X. and Y.Y.; methodology, X.X.; software, X.X.; validation, X.X., Y.Y. and Z.Z.; formal analysis, X.X.; investigation, X.X.; resources, X.X.; data curation, X.X.; writing—original draft preparation, X.X.; writing—review and editing, X.X.; visualization, X.X.; supervision, X.X.; project administration, X.X.; funding acquisition, Y.Y. All authors have read and agreed to the published version of the manuscript.” Please turn to the CRediT taxonomy for the term explanation. Authorship must be limited to those who have contributed substantially to the work reported.
Funding: OK
Data Availability Statement: In this section, please provide details regarding where data sup-porting reported results can be found, including links to publicly archived datasets analyzed or generated during the study. Please refer to suggested Data Availability Statements in section “MDPI Research Data Policies” at https://www.mdpi.com/ethics. You might choose to exclude this statement if the study did not report any data.
Acknowledgments: In this section, you can acknowledge any support given which is not covered by the author contribution or funding sections. This may include administrative and technical support, or donations in kind (e.g., materials used for experiments).
Conflicts of Interest: Declare conflicts of interest or state “The authors declare no conflict of interest.” Authors must identify and declare any personal circumstances or interest that may be perceived as inappropriately influencing the representation or interpretation of reported research results. Any role of the funders in the design of the study; in the collection, analyses or interpretation of data; in the writing of the manuscript, or in the decision to publish the results must be declared in this section. If there is no role, please state “The funders had no role in the design of the study; in the collection, analyses, or interpretation of data; in the writing of the manuscript, or in the decision to publish the results”.
The references aren’t in format. Please verify all references and used guide for authors.

Reviewer 2 Report
Manuscript entitled “Chitosan/phosphate rock-derived natural polymeric composite to sequester divalent copper ions from water” submitted by Rachid El Kaim Billah, Moonis Ali Khan, Saikh Mohammad Wabaidur, Byong-Hun Jeon, Amira Am, Hicham Majdoubi, Younesse Haddaji, Mahfoud Agunaou, Abdessadik Soufiane, can be accepted for publication in Nanomaterials Journal, in this form. The objectives of this study are presented from the beginning and followed closely throughout the manuscript. The experimental data are clearly presented and discussed in detail. However, I have one observation:
- Page 2, line 32: Herein, we synthesized amine (-NH2), hydroxyl (-OH) and phosphate (PO4 3-) groups enriched chitosan (CH)”. This paragraph should be reformulated.
Reviewer 3 Report
The paper presents the studies on chitosan/phosphate rock-derived natural polymeric composite to sequester divalent copper ions from water. The presentation of methods and scientific results in the current form is not satisfactory for publication in the Nanomaterials journal. The minor and major drawbacks to be addressed can be specified as follows:
1. Page 7, lines 158-160, “Brunauer–Emmett–Teller (BET, ASAP 2020, 158 Micromeritics, Norcross, USA) surface area analyzer” - not a very happy description. In my opinion it should be better this way: “ASAP 2020 (Micromeritics, Norcross, USA) surface area analyzer was employed to measure the surface area (the BET (Brunauer–Emmett–Teller) method)”.
2. Fig1, Fig. 2, and others. I would consider the order in which the data are presented: CH, TNP, CH-TNP. There are different sequences of presented data in the individual figures and their placement in the legend.
3. Page 11, line 240. How did the authors notice the micropores in Fig. 2 at a resolution of 100 micrometers?
4. Page 11, line 242. Fap?
5. Page 11, line 245. Would you please provide the experimental result(s) of the measurement confirming the monolayer formation?
6. Page 11, lines 246-255. Would you please comment on the value of the maximum adsorption (the total pore volume) and the values of the hysteresis loop area.
7. Fig. 2, legend. CH, CH-TNP, TNP ---> CH, TNP, CH-TNP.
8. Page 13, Tab. 1. (i) How was the pore width calculated? (ii) Pore size ---> Average pore size.
9. At what temperature and how long were the samples desorbed (outgassed) before the nitrogen adsorption measurement?
10. Fig. 3. Sloppy drawing - different font sizes. A different sequence of sample names in legends.
11. Fig. 2. Why no data for TNP?
12. Fig. 4. An unnecessary frame on A panel.
13. Page 17, line 364, “were 32 and 44.84 mg/g”. See Tabs. 2 and 3. 37.73? 45.87?
14. See all the manuscript. J/mol-K ---> J/mol/K (e.g. Tab. 5). HCL ---> HCl (e.g. Fig. 5). NaCL ---> NaCl (e.g. Fig. 5).
15. Fig. 5. (i) Cycle Number ---> Cycle number. (ii) commas should be replaced with dots (e.g. 81,6 ---> 81.6).
16. Page 5, Conclusions. There is nothing in the conclusions about Fig. 6.
17. I suggest authors use XPS instead of FTIR.
Round 2
Reviewer 1 Report
The paper “Chitosan/phosphate rock-derived natural polymeric composite to 1 sequester divalent copper ions from water”
authors: Rachid El Kaim Billah, Moonis Ali Khan*, Saikh Mohammad Wabaidur, Byong-Hun Jeon, Amira AM, Hicham Majdoubi, Younesse Haddaji, Mahfoud Agunaou, Abdessadik Soufiane, presents an interesting topic to Nanomaterials readers, but further corrections are necessary for its acceptance.
My principal questions or remarks:
The title is clear. The content is in accord with title.
The manuscript adheres to the journal's standards after revision.
The size of the article is appropriate to the contents.
The authors must underline the major findings of their work and explain how the use of their proposed procedures and materials represent a progress to other similar studies. In response the authors justified the novelty, but in manuscript these aspect are not clear.
The Abstract was revised.
The key words permit found article in the current registers or indexes.
In the introduction it is clearly described the state of the art of the investigated problem.
The methods are well described and the equipment and materials have been adequately described.
The tables contain necessary results.
The figures have a good quality.
The Conclusion is been justified sufficiently.
The literature is sufficiently critical, current, and internationally evaluated.
The paper has the text presented and arranged clearly and concisely.
The English was corrected.
PLEASE USE TEMPLATE FOR MANUSCRIPT. On the site is template:
https://www.mdpi.com/journal/nanomaterials/instructions
Please respect author guide.
Author Contributions: The following statements should be used “Conceptualization, X.X. and Y.Y.; methodology, X.X.; software, X.X.; validation, X.X., Y.Y. and Z.Z.; formal analysis, X.X.; investigation, X.X.; resources, X.X.; data curation, X.X.; writing—original draft preparation, X.X.; writing—review and editing, X.X.; visualization, X.X.; supervision, X.X.; project administration, X.X.; funding acquisition, Y.Y. All authors have read and agreed to the published version of the manuscript.” Please turn to the CRediT taxonomy for the term explanation. Authorship must be limited to those who have contributed substantially to the work reported.
Funding: OK
Data Availability Statement: In this section, please provide details regarding where data sup-porting reported results can be found, including links to publicly archived datasets analyzed or generated during the study. Please refer to suggested Data Availability Statements in section “MDPI Research Data Policies” at https://www.mdpi.com/ethics. You might choose to exclude this statement if the study did not report any data.
Acknowledgments: In this section, you can acknowledge any support given which is not covered by the author contribution or funding sections. This may include administrative and technical support, or donations in kind (e.g., materials used for experiments).
Conflicts of Interest: Declare conflicts of interest or state “The authors declare no conflict of interest.” Authors must identify and declare any personal circumstances or interest that may be perceived as inappropriately influencing the representation or interpretation of reported research results. Any role of the funders in the design of the study; in the collection, analyses or interpretation of data; in the writing of the manuscript, or in the decision to publish the results must be declared in this section. If there is no role, please state “The funders had no role in the design of the study; in the collection, analyses, or interpretation of data; in the writing of the manuscript, or in the decision to publish the results”.
Reviewer 3 Report
Congratulations on a great job. The author has made a substantial improvement for this article. However, I have a few more comments.
1. In Fig. 1 it should be the following order (from top to bottom) CH, TNP, CH-TNP (as shown in the other figures).
2. The font size should be standardized in Fig. 3
3. My commentsŁ: “At what temperature and how long were the samples desorbed (outgassed) before the nitrogen adsorption measurement?”
Response to comment: “Before the nitrogen adsorption measurement, samples were outgassed at 150°C for 18h.”
What substances were removed during degassing? Water? The studied sample? Looking at the curves in Fig. 3A; unfortunately, I think the samples. This puts the reliability of the nitrogen adsorption isotherm measurements in a bad light for these samples. A methodological error has crept in here. It is also worth emphasizing that the selection of the sample degassing temperature, in this case, is not an easy task.
Round 3
Reviewer 3 Report
The authors have made a substantial improvement for this article. The manuscript can be accepted for publishment in the present form.